# Pathophysiological Mechanisms by which Heat Stress Potentially Induces Kidney Inflammation and Chronic Kidney Disease in Sugarcane Workers

**DOI:** 10.3390/nu12061639

**Published:** 2020-06-02

**Authors:** Erik Hansson, Jason Glaser, Kristina Jakobsson, Ilana Weiss, Catarina Wesseling, Rebekah A. I. Lucas, Jason Lee Kai Wei, Ulf Ekström, Julia Wijkström, Theo Bodin, Richard J. Johnson, David H. Wegman

**Affiliations:** 1School of Public Health and Community Medicine, Institute of Medicine, Sahlgrenska Academy, University of Gothenburg, Box 414, 405 30 Gothenburg, Sweden; kristina.jakobsson@amm.gu.se; 2La Isla Network, 1441 L Street NW, Washington, DC 20005, USA; jason@laislanetwork.org (J.G.); ilana@laislanetwork.org (I.W.); ineke@laislanetwork.org (C.W.); r.a.i.lucas@bham.ac.uk (R.A.I.L.); ulf.ekstrom@med.lu.se (U.E.); david_wegman@uml.edu (D.H.W.); 3Faculty of Epidemiology and Population Health, London School of Hygiene and Tropical Medicine, Keppel Street, London WC1E 7HT, UK; 4Occupational and Environmental Medicine, Sahlgrenska University Hospital, Box 414, 405 30 Gothenburg, Sweden; 5Institute of Environmental Medicine, Karolinska Institutet, Nobels väg 13, 171 65 Solna, Sweden; theo.bodin@ki.se; 6School of Sport, Exercise & Rehabilitation Sciences, University of Birmingham, 142 Edgbaston Park Rd, Birmingham B15 2TT, UK; 7Department of Physiology, Yong Loo Lin School of Medicine, National University of Singapore, 2 Medical Drive, MD9, National University of Singapore, Singapore 117593, Singapore; phsjlkw@nus.edu.sg; 8Global Asia Institute, National University of Singapore, 10 Lower Kent Ridge Rd, Singapore 119076, Singapore; 9N.1 Institute for Health, National University of Singapore, 28 Medical Dr, Singapore 117456, Singapore; 10Department of Laboratory Medicine, Division of Clinical Chemistry and Pharmacology, Lund University, 221 85 Lund, Sweden; 11Division of Renal Medicine, Department of Clinical Science Intervention and Technology, Karolinska Institutet, 141 86 Stockholm, Sweden; julia.wijkstrom@ki.se; 12Division of Renal Diseases and Hypertension, School of Medicine, University of Colorado Denver, Aurora, CO 80045, USA; richard.johnson@cuanschutz.edu; 13Department of Work Environment, University of Massachusetts Lowell, Lowell, MA 01845, USA

**Keywords:** kidney, heat, acute kidney injury, occupation, inflammation, hydration, heat stress

## Abstract

Background: Chronic kidney disease of non-traditional origin (CKDnt) is common among Mesoamerican sugarcane workers. Recurrent heat stress and dehydration is a leading hypothesis. Evidence indicate a key role of inflammation. Methods: Starting in sports and heat pathophysiology literature, we develop a theoretical framework of how strenuous work in heat could induce kidney inflammation. We describe the release of pro-inflammatory substances from a leaky gut and/or injured muscle, alone or in combination with tubular fructose and uric acid, aggravation by reduced renal blood flow and increased tubular metabolic demands. Then, we analyze longitudinal data from >800 sugarcane cutters followed across harvest and review the CKDnt literature to assess empirical support of the theoretical framework. Results: Inflammation (CRP elevation and fever) and hyperuricemia was tightly linked to kidney injury. Rehydrating with sugary liquids and NSAID intake increased the risk of kidney injury, whereas electrolyte solution consumption was protective. Hypokalemia and hypomagnesemia were associated with kidney injury. Discussion: Heat stress, muscle injury, reduced renal blood flow and fructose metabolism may induce kidney inflammation, the successful resolution of which may be impaired by daily repeating pro-inflammatory triggers. We outline further descriptive, experimental and intervention studies addressing the factors identified in this study.

## 1. Introduction

The past decades’ temperature increase has coincided with the recognition of high levels of chronic kidney disease unrelated to traditional risk factors such as diabetes or hypertension among heavy laborers in disadvantaged communities around the equator [1,2,3]. The condition is referred to as Chronic Kidney Disease of non-traditional origin (CKDnt) [4], though other terms have also been used to differentiate the condition from common CKD (Uncertain Etiology (CKDu), or, in Central America, Mesoamerican nephropathy [5]). There is increasing evidence that strenuous physical activity generating high metabolic heat when combined with high external heat exposure is associated with a risk for decreased kidney function and injury in populations at risk of CKDnt [6,7,8].

In Mesoamerica, CKDnt is predominantly seen in hot lowland regions [9] where sugarcane is cultivated [10,11,12]. Substantial decreases in renal function at a magnitude comparable to acute kidney injury (AKI) criteria [13]; hereafter denoted incident kidney injury (IKI), is more often seen in manual cane cutters during the five-month harvest than other workers performing less strenuous work in the same environment [6]. Apart from sugarcane workers [14,15,16,17,18,19,20,21,22], other heat-exposed workers also develop AKI-like conditions, such as outdoor agricultural workers without access to shade [23], brick-kiln workers [24] and miners [25], suggesting heat-related kidney injury is generalizable to strenuous manual work in hot environments. Of note, high rates of AKI have been reported in marathon runners [26,27,28], though such injury seems transient with no reports of long-term consequences [29]. However, an important point of difference between athlete and worker is that an athlete’s training and race schedule allows for recovery and changes in workload unlike many workers who undertake strenuous manual labor.

Reduced kidney function in populations at risk of CKDnt has often been shown to coincide with signs and/or symptoms of inflammation such as fever, elevated C-reactive protein (CRP), leukocytosis, or leukocyturia [6,14,15,23,29,30,31]. Kidney biopsies performed in workers with acute kidney injury have also shown inflammation [29]. This inflammatory process has been hypothesized as a possible mediator of heat-induced kidney injury [32] or the result of a toxin or infection [31,33]. 

We have previously reported a strong association between workload and cross-harvest kidney function reduction/injury, self-reported fever, and moderately elevated circulating CRP levels in manual sugarcane cutters [6]. CRP levels in cane cutters with kidney injury were higher than levels typically seen in populations with worse kidney function [34,35]; thus, it is unlikely that reduced kidney function per se causes the CRP elevation. Rather, increased CRP levels seem associated with a decrease in kidney function, pointing to either a systemic inflammatory condition causing kidney injury, systemic inflammation induced by recent kidney injury from other factor/s, or a process causing both systemic inflammation and kidney injury in parallel. 

Schlader et al. recently published a review on how physical work in heat could lead to kidney injury [36]. The aim of our paper is to supplement that review using a perspective that accounts for the systemic inflammation as well. We present original data in support of this perspective and consider what these findings imply for further practice and research. This paper is structured as follows: 

First, we develop our theoretical framework (Figure 1), which largely originates in heat and exercise pathophysiology literature.

Then, using this framework we report empirical findings in sugarcane workers to explore if data collected in Mesoamerica support an association between IKI and indicators of proposed pathways of kidney damage, while identifying knowledge gaps. Indicators of these pathways explored include consumption of sugary drinks, water and electrolyte solutions, use of non-steroidal anti-inflammatory drugs (NSAID), as well as self-reported fever and markers of inflammation, uric acid metabolism, rhabdomyolysis, anemia, and electrolyte disturbances.

Finally, we consider what implications the findings have for research and preventive efforts.

## 2. The Theoretical Framework

The framework encompasses systemic and local inflammation triggers co-occurring with renal blood flow and metabolism mismatch. This framework applies to the reality for many manual sugarcane workers in Mesoamerica: occupational exposure (6–8 h for 6 days per week) to strenuous physical activity (~55% of maximum heart rate) in heat (~30 °C wet-bulb globe temperature). The work intensity is driven by piecework payment, incentivizing maximized performance over a five-month period. 

### 2.1. Systemic and Kidney Inflammation Triggers

Elevated levels of some cytokines have been associated with kidney injury among marathon runners [27,28] and research participants following strenuous exercise in heat [37]. Systemic inflammation is a frequent finding immediately after strenuous exercise, although markers of systemic inflammation typically decrease with increasing levels of regular low-moderate physical activity [38]. However, repetitive prolonged excessive training without sufficient recovery is hypothesized to disturb the resolution of the acute inflammatory reaction, leading to chronic, pathological systemic inflammation [39].

A number of overlapping and likely interacting mechanisms that separately or together may ignite an inflammatory reaction in the kidney and/or systemically are outlined below:

#### 2.1.1. Gut Permeability, Endotoxins, and Cytokines

Hard and sustained physical exercise can lead to decreased gut perfusion and increased gut permeability [40,41,42,43,44,45,46], a process potentially aggravated by elevated body core temperature from exogenous heat stress [45,46,47,48]. As gut permeability increases, bacterial endotoxins may enter the bloodstream and trigger the release of pro-inflammatory cytokines to the point of multi-organ failure and altered mental status as observed in exertional heat stroke [49,50,51,52], or more limited organ damage as seen in heat injury [51,52]. Repeated clinical or subclinical heat stroke events have previously been considered as a cause of CKDnt [53,54]. Recently there has been considerable interest in the effects of systemic inflammation on kidney disease. For example:

(A) The understanding of AKI in sepsis has been moving from a perfusion to an “inflammatory-toxic” theory, with endotoxin effects on tubular cells combined with cytokine-induced tubular apoptosis identified as potential mechanisms [55]; 

(B) Gut–kidney crosstalk with decreased kidney function increasing gut permeability has been hypothesized to fuel systemic inflammation and thereby progressive renal damage both as a cause of CKD progression [56,57] and AKI in septic shock [58]; and 

(C) Longitudinal studies have found inflammatory markers that can predict kidney function loss [59,60,61]. 

In rodent models, kidneys with acute injury from various causes were more prone to a strong inflammatory response to endotoxin [61,62,63,64] and previous endotoxin exposure aggravated the kidney inflammatory response to subsequent repeated endotoxin exposure [65]. Thus, it seems that endotoxins may aggravate kidney injury from other causes and that there is little adaptation to repeated endotoxemia.

#### 2.1.2. Muscle Cellular Breakdown Byproducts and Cytokines

While exertional heat stroke is often accompanied by rhabdomyolysis-level creatine phosphokinase (CPK) elevation [52] such pronounced elevations were not seen in exertional heat stroke cases with isolated kidney rather than multi-organ failure [66]. In an experimental setting, muscle-damaging exercise that mildly elevated CPK levels was identified as a risk factor for developing AKI during strenuous physical exercise in heat [37]. From this it could be suggested that non-rhabdomyolysis-level muscular damage may also contribute to kidney injury in heat-stressed workers, perhaps through muscle breakdown releasing uric acid [67] or muscle release of cytokines [37], promoting a systemic inflammation with subsequent effects on the kidney.

#### 2.1.3. Sugar Intake, Fructose and Uric Acid

The combination of dehydration, heat and rehydration with soft drink-like fluids has been studied experimentally in mice [68,69] and humans [70], finding that sugary drink intake exacerbates kidney injury. Rehydration with sugary drinks could induce kidney inflammation as tubular fructose metabolism stimulates reactive oxygen species and chemokine secretion [71] along with local uric acid production [68]. Tubular uric acid, both crystalline and non-crystalline, is pro-inflammatory [67]. 

### 2.2. Kidney Blood Flow and Metabolism Mismatch

Dehydration and hyperthermia are two distinct [72,73], though often conflated [72,74] and co-incident [72], consequences of excessive heat exposure. As with strenuous physical exercise [75,76,77], these independently [47,78] lead to reduced renal blood flow (RBF) as blood is redirected to the working muscle and to the skin to dissipate heat. This redistribution is regulated by several mechanisms: 

(A) Activation of the sympathetic nervous system (SNS) decreases RBF. Both strenuous exercise [79] and hyperthermia [80] increase SNS activity. Notably, agricultural work often involves whole body exercise, which possibly elicits a stronger SNS response than lower-body oriented exercise such as running and cycling [79]. 

(B) The renin-angiotensin-aldosterone-(RAAS)-vasopressin system has the potential to decrease RBF while increasing tubular metabolism. The RAAS is activated by electrolyte and fluid loss from sweating, reducing RBF and increasing tubular water and sodium reabsorption [81]. Vasopressin, released with RAAS activity and fructose consumption [68,69,70,82,83], reduces RBF and promotes sodium reabsorption. Increased sodium and water reabsorption increase renal energy and oxygen demand [84]. 

(C) Loss of potassium through RAAS activation and/or tubular injury may have secondary effects on renal blood flow. Long-term potassium depletion may lead to tubulointerstitial inflammation, a condition called hypokalemic or kaliopenic nephropathy [85,86]. Although this condition is not well studied in humans [85,86], it is related to vasoconstriction and impaired angiogenesis in rodents [87].

(D) Prostaglandin increases RBF by relaxation of the afferent arteriole, a safety mechanism at low renal perfusion, which may be blunted by NSAIDs inhibiting prostaglandin synthesis [88,89]. NSAIDs, often used among agricultural workers in Nicaragua [6,90], have been shown to reduce RBF in exercising humans [91].

(E) Endothelial nitric oxide, the synthesis of which may be inhibited by uric acid [92], increases RBF. Uric acid is often elevated after strenuous physical activity [93,94,95], possibly secondary to muscle injury [67]. 

Reduced renal blood flow translates to reduced removal of waste products and reduced oxygen delivery, the latter which may be further impaired by anemia. Red blood cell synthesis is stimulated by erythropoietin, a hormone synthesized by fibroblasts located in the kidney tubuli. It has been suggested that injury to these cells may lead to decreased erythropoietin synthesis, decreasing hematocrit [96].

In summary, the proposed framework encompasses the range of risk factors and mechanisms that may have an unfavorable effect on kidney health in repeatedly heat-stressed workers. The combination of dehydration, muscle injury, strenuous exercise, exogenous heat exposure, and musculoskeletal pain treated by over-the-counter medications, and anemia, thus include factors that potentially reduce RBF and oxygenation, while also increasing metabolic demands on the tubular cells. The combination of some/all of these factors may tilt the tubular oxygen balance towards hypoxia, which in itself may evoke inflammation [89,97]. The combined effect of hypoxia and other pro-inflammatory stimuli may further aggravate injury. 

## 3. Empirical Findings in Sugarcane Workers

We provide evidence in support of the theoretical framework using a combination of original data from two sugarcane worker cohorts, each covering two seasons: one set collected in 2014–2016 at Ingenio El Angel (IEA), El Salvador [17,96], and the other set collected in 2017–2019 at Ingenio San Antonio (ISA), Nicaragua, the latter part of the Adelante Initiative Intervention Study [6]. Our own data are presented in the context of published results from other occupational studies of CKDnt in Mesoamerica [7].

### 3.1. Material and Methods

#### 3.1.1. Own Data

##### Overview

The designs of the IEA and Adelante cohorts have been described previously [6,8,18,98]. Briefly, workers answered a questionnaire and had serum and urine samples collected before start of harvest and towards the end of harvest and, for the IEA cohort, also twice during harvest. Liquid intake was estimated by asking the worker to recall how many containers of specific sizes and fluid types were consumed the previous day. Appendix A provides a basic description of the second year IEA cohort, the results of which have not been reported elsewhere. In both cohorts, workers with no follow-up after baseline were excluded. In the IEA cohort, which also had post-shift measurements, only pre-shift measurements were considered. For the Adelante study, which included non-cutter sugarcane workers, we only considered seed and burned cane cutters as these are the jobs for which the vast majority of IKI cases have been recorded [6,8].

All analyses were sex stratified, but due to the very low numbers of IKI among women (total of N = 6, 5 of whom are from the second year of IEA), we can only present data for women from second-year IEA (Appendix A). 

Important differences between the populations in these studies include the use of a pre-employment creatinine cut-off at ISA but not at IEA. Consequently, the baseline kidney function at IEA was much lower and the workforce significantly older than at ISA (Table 1).

##### Laboratory Procedures

SCr, CRP, CPK, sodium, magnesium, potassium, and uric acid were measured on a Cobas 701 instrument (Roche Diagnostics, Basel, Switzerland) at the Department of Clinical Chemistry, Skåne University Hospital in Lund, Sweden [6,18]. Magnesium and potassium were only analyzed for the samples from El Salvador. Urine dipstick (Bayer 10-SG MultiStix Urine Dipsticks analyzed using Siemens CLINITEK Status®+ Analyzer) and microcopy were analyzed at Centro de Hemodiálisis, San Salvador, El Salvador and blood counts were analyzed at Laboratorio CECIAM Escalón, San Salvador, El Salvador [18]. As previously described, some samples in year 2 El Salvador and year 1 Nicaragua were inadequately mixed after thawing, and a validated sodium-based correction factor was used to correct the concentrations in these years [6,99], meaning sodium results from these years cannot be used. 

##### Outcome Definition

Incident kidney injury: SCr increase at follow-up ≥0.30 mg/dL higher than or ≥1.5-times the baseline value (IKI_Crea_) [6].

##### Covariate Definitions

eGFR was calculated using the CKDEPI equation [100] for creatinine, assuming non-black race, and categorized to <45, 45–60, 60–90 and >90 mL/min/1.73 m^2^ based on established categories for classifying chronic kidney disease [101]. Age was categorized to 18–30, 30–40, 40–50 and >50 years. C-reactive protein (CRP) was categorized to <3, 3–10, 10–20 and >20 mg/L, sodium to <137 (hyponatremia), 137–145 and >145 mmol/L (hypernatremia), potassium to above or below 3.5 mmol/L (hypokalemia), and magnesium to above or below 0.7 mmol/L (hypomagnesemia), based on reference intervals at the laboratory of analysis, Clinical Chemistry at Lund University Hospital, Sweden. The CRP cut-offs above 3.0 mg/L were based on the distribution of values and from clinical experience of interpretation of low-moderate CRP levels. Magnesium was only measured at baseline. Incident anemia was defined as hemoglobin at <125 g/L at follow-up while above this at baseline. The 125 g/L hemoglobin threshold was set as the mean baseline hemoglobin –2 standard deviations. Incident hematuria, proteinuria, and presence of urate crystals or non-hyaline cylinders (granular or leukocyte casts) or pronounced leukocyturia in sediment which was defined as such abnormality not present at baseline. Hematuria and proteinuria were assessed using dipsticks. Workers with any of these abnormalities at baseline were excluded from the analysis of the corresponding incident abnormality as per definition such abnormality was not incident. Similarly, workers with fever within the past week(s) (1 week in year 1 Nicaragua, 2 weeks in year 2 Nicaragua) who reported fever at baseline were excluded from these analyses.

Consumption of *bolis* was analyzed in the Adelante cohort. *Bolis* are 300 mL electrolyte solution bags reported almost exclusively as consumed during work (between breakfast and noon), the period during which most water consumption is also reported. *Bolis* are produced by the mill and contain 7 grams of sugar per dl, 50 mg of sodium chloride and 20 mg of potassium phosphate. The water drunk during work is supplied from the mill’s tested purification facility and also used for *boli* production [102]. We categorized the volume of *boli* and water consumption reported between breakfast and noon to 0, 300–900 mL and >900 mL and 0–2 L, >2–5 L and >5 L, respectively. Liquid intake categories were defined to identify those with no or very low intake as a separate subgroup, and then the remainders were divided into two groups using a rounded figure that gave similarly sized groups. 

##### Statistical Analysis

Categorical Covariates

El Salvador: Mixed-effects logistic regression with random effect for each worker was used to estimate odds ratios (OR) with 95% CI for IKI at a specific follow-up day during harvest for each of the categorical explanatory variables at that same day. ORs were estimated with and without baseline eGFR adjustment.

For categorical covariates for which there were too few workers with an abnormal covariate value to permit mixed-effects logistic regression of IKI (covariates assessed only in year 1 IEA: dipstick, whole blood indices, urine microscopy), we used mixed linear regression of eGFR instead.

Nicaragua: Incidence ratios (IRs) with 95% confidence intervals (CI) were calculated for the explanatory variable categories, including data from both years modelled using Poisson regression with a random effects structure accounting for repeated observations within the same individual. IRs were estimated with and without baseline eGFR adjustment. We also performed multivariate analyses to adjust for confounding due to job and calendar year (harvest 1 vs. 2), and to assess confounding between NSAID, fever and CRP and liquid intake types respectively (Appendix A).

Continuous Covariates

Median and interquartile range of blood white blood cell (WBC), monocyte, lymphocyte and neutrophil count, and CRP, creatine phospho-kinase (CPK), potassium, sodium, hemoglobin and uric acid levels were calculated for all at baseline, and separately for those with and without IKI at end-harvest (Nicaragua) and at follow-up visit (El Salvador).

Difference in these parameters between IKI and non-IKI on follow-up days (El Salvador) and between baseline and follow-up at end-harvest for IKI and non-IKI workers (Nicaragua) was estimated using mixed effects linear regression. Each worker had his own random effect to account for repeated measurements within the same individual.

Reverse causation, with a decrease in kidney function potentially leading to increased serum concentrations of explanatory biomarkers, was addressed by adjusting for current eGFR. Tobit rather than linear regression was used for CRP in Nicaraguan cohort as CRP was truncated at 0.6 mg/L with levels below that not quantified.

#### 3.1.2. Ethical Approval

This study was approved by the Comite de Etica para Investigaciones Biomedicas (CEIB), Facultad de Ciencias Medicas, Universidad Nacional Autonoma de Nicaragua (UNAN—Leon (FWA00004523/IRB00003342) and the The National Ethics Committee for Clinical Research (Comité Nacional de Ética de Investigación de Salud) of El Salvador (OHRP IRB No. 0005660, FWA No. 00010986). The biochemical investigations carried out at the Division of Clinical Chemistry and Pharmacology at Lund University in Sweden were approved by the Regional Ethical Review Board in Lund (reg no 2018-256 and 2016-60).

#### 3.1.3. Other Published Data

We relied on a concise literature review [7] of epidemiological data published on CKDnt in Central America.

## 4. Results with Discussion

### 4.1. Inflammation Coinciding with Kidney Injury

Several studies have noted self-reported fever and objective measures of inflammation (in our case CRP) were associated with reduced kidney function indicating incident kidney injury (IKI) (Table 1 and Table 2) [6,14,23,29,30,31,99]. The co-occurrence of non-hyaline casts, leukocyturia, hematuria, and proteinuria plus systemic markers of inflammation in this study (Table 1) and a previous study [29] suggests the kidney as a/the locus of inflammation. 

Although hyperthermia has been recorded in sugarcane workers with recent kidney injury [31], ’fever’ as a symptom is a more complex phenomenon influenced by cultural factors and understanding of illness. In the second year of the Adelante study, when workers reporting fever during harvest were asked what they attributed their most recent fever to, 50% answered sun exposure when given the pre-specified alternatives flu, other infection, sun exposure, and an open-ended alternative [8]. 

Of the workers with a urine dipstick examination none had a positive nitrite test. While this negative finding indicates that urinary tract infections by gram-negative bacteria are unlikely, we cannot present any new data to rule in or out infectious causes of inflammation. Previous occupational studies from Mesoamerica have had negative findings for urine cultures [30,102], leptospirosis and hantavirus serology [103,104,105], and non-specific DNA and RNA screening for pathogens in urine, serum, and kidney tissue [103]. Infection increases the risk of concomitant heat injury [51,52], meaning a potential causal role of infections is still consistent with our heat hypothesis. As many workers affected by CKDnt are paid via piece rate and not or only partially compensated when off work due to illness, there are incentives to work when feeling unwell or not completely recovered from infections.

### 4.2. Systemic and Kidney Inflammation Triggers

#### 4.2.1. Gut Permeability, Endotoxins, and Cytokines

Increased intestinal permeability has been seen experimentally in humans at body core temperatures (slightly above 38 °C), especially when combined with NSAID consumption [106]. Similar body core temperatures have been recorded and estimated in sugarcane workers (Rebekah Lucas, forthcoming). Endotoxemia from intestinal leakage [107,108] and increased cytokine levels [107] have been shown in multi-day ultra-endurance athletes, an activity of similar intensity to sugarcane cutting [109], but no study has reported endotoxins or plasma cytokine levels in such activities sustained over months, neither in athletes nor in sugarcane cutters. As such there is neither current evidence in favor nor against such a hypothesis. Biopsy findings in kidneys of heat stroke victims [25,51] and mice exposed to pre-heat stroke heat strain [110] and endotoxins [111,112], however, show tubulointerstitial inflammation, just as sugarcane workers with AKI [29].

#### 4.2.2. Muscle Cellular Breakdown Byproducts and Cytokines

There were no cases of a rhabdomyolysis-level CPK (>5-times above upper limit of reference interval) in our cohorts or in sugarcane workers with cross-shift kidney injury [15]. In our cohorts, CPK increased during harvest and was slightly positively associated with IKI in El Salvador (Table 2). Preliminary reports suggest sugarcane cutters lose muscle mass during harvest [113], also indicating their level of physical activity may lead to muscular damage. Considering muscle-damaging exercise has been found to increase sensitivity to heat-induced kidney injury [37], it is plausible that sub-rhabdomyolysis muscle damage contributes to kidney injury among sugarcane cutters. 

#### 4.2.3. Sugar Intake, Fructose, and Uric Acid

The increased risk of kidney injury with sugary liquid intake among cutters (Table 1) supports a role of tubular fructose and uric acid metabolism. Serum uric acid increase was linked to eGFR decrease in both cohorts (Table 2), something also seen in other studies [15,22]. We saw larger eGFR decreases on days urinary uric acid crystals and low U-pH were detected (Table 1 and Table 2), potentially supporting a role of cyclic uricosuria [67]. While biopsy studies have not identified uric acid crystals [29,114], it should be noted that biopsies were taken from those not currently working, and such crystals are likely to be transient. Cross-sectional studies have reported high serum uric acid levels in CKDnt [17,115,116] and possible associations with sugar intake during work [117]. Three studies have examined urinary uric acid levels among cutters [15,19,67], one finding a cross-shift decrease [15], and another an increase which may be larger during the hottest days [19]. More studies are needed to determine the relation between uricosuria and kidney injury.

### 4.3. Kidney Blood Flow and Metabolism Mismatch

The increased risk with NSAID use observed previously and in our research, at least in combination with dehydration [14], suggest RBF reductions may be important. Notably, the corticomedullary junction is where most inflammation is seen in biopsies of sugarcane workers with recent kidney injury [29]. The outer medulla is the part of the kidney most sensitive to hypoxia, owing to its large metabolic demand and low blood flow [118], putting it at the “brink of hypoxia” [89]. 

Anemia was common in CKDnt cases [17,115], and has coincided with “acute” CKDnt [30,31] and IKI (Table 1) in sugarcane workers. Even when minimal, reduced hematocrit may further impair oxygen delivery to tubular cells at the margin of hypoxia. In order to further understanding of the role of anemia studies of its underlying cause; systemic inflammation, kidney injury, and/or several other potential mechanisms are warranted. In the IEA cohort, the high mean corpuscular volume (MCV) of the cases of incident anemia (median MCV 96 fL, range 89–103 fL) indicate that iron deficiency anemia does not seem to be the primary explanation. Measuring erythropoietin levels may inform whether anemia is secondary to tubular injury. 

#### Renin–Angiotensin–Aldosterone System, Vasopressin Activation, and Hypokalemia

In our data, hypokalemia was strongly associated with IKI (Table 1). Hypokalemia was seen in most sugarcane workers with “acute” CKDnt [30], frequently developed across sugarcane work shifts [17] and is common in CKDnt patients [114,119,120]. Others however have found normal potassium levels in sugarcane workers with kidney injury [22]. In the IEA cohort, hypomagnesemia and hypokalemia at baseline predicted IKI, and follow-up hypokalemia was associated with IKI. These associations however seem partly explained by pre-existing kidney injury as baseline eGFR-adjustment explained all or large parts of these associations. Such electrolyte disturbances may be signs of manifest disease and differences between studies may be due to including populations at different stages of disease. 

Few workers developed hyponatremia (<137 mmol/L) during harvest, and it occurred, it was mild (one worker at 133 mmol/L, the rest at ≥135 mmol/L). The absence of sodium disturbances in the morning indicate that workers’ hormonal systems and kidney structures primarily responsible for sodium and volume regulation (RAAS/vasopressin and tubuli, respectively) were functional enough to maintain sodium homeostasis by the morning after a workday wherein approximately 7 L/work shift of water [8] was drunk. Our results contrast with previous reports of hyponatremia being common in active sugarcane workers [15,22]. One of these studies [15] reported post-shift sodium levels and had a strong focus on increasing liquid intake by instructing workers to drink 16 L of water per work shift. Such instructions need to be given with much caution in order not to induce potentially fatal hyponatremia [72]. 

Electrolyte solutions are commonly recommended for rehydration during exercise in heat. In the Nicaraguan cohort, a suggested protective effect of *boli* intake seems due to their electrolyte rather than fluid or sugar content as the *boli* consumption was quantitatively small compared to water intake and sugary liquid intake was detrimental (Table 1). Findings from cross-sectional studies on *boli* consumption have been mixed [114,117], but one cross-shift study found a protective effect of electrolyte consumption on cross-shift AKI [11] and one study found a protective effect on cross-harvest eGFR decline [21]. Considering the high occurrence of hypokalemia in workers with recent kidney injury and the protective effects of electrolyte supplementation, a potential role of hypokalemia as contributing to CKDnt progression and not just as an indicator of tubular damage and RAAS activation may need further consideration. We have found no study assessing the electrolyte content of sugarcane worker diets and it is likely that the *boli* electrolyte content differs between companies.

## 5. Summary and Implications

Inflammation biomarkers and fever were associated with kidney injury among sugarcane cutters, an observation that could provide insight into pathogenesis of kidney damage. However, we argue that signs of inflammation in kidney biopsies and serum, rather than pointing to an infection or toxin/precipitant as the cause of injury [31,33], are indicators of heat-induced kidney injury, which could also be expected to be mediated by inflammation. We postulate that kidneys of sugarcane cutters are potentially exposed to pro-inflammatory stimuli including hypoxia, fructose, uric acid and circulating endotoxins and cytokines (Figure 1) due to strenuous physical activity in heat with concomitant sugar consumption.

The frequency with which a prolonged exposure to heat and exercise occur is one potentially crucial difference between the settings in which AKI has been noted in athletes and sugarcane cutters. Sugarcane cutters repeat the same activity for six or sometimes even seven days a week for five–six consecutive months, meaning the possibilities for recuperative rest are more limited than for athletes. Rest is considered beneficial for the healing of an inflamed heart [121,122] and skeletal muscles [39]. It is a reasonable conjecture that rest is necessary for adequate healing of inflamed kidneys as well, and that differences in rest taken after AKI between piece-paid workers in developing countries and athletes could explain the different long-term outcome after heat/exercise-induced kidney injury. Pushing kidneys to the physiological limits daily, causing high tubular reabsorption demand while reducing blood flow, and repeating pro-inflammatory stimuli, may contribute to fibrotic changes rather than successful healing. 

### 5.1. Implications for Research

A better understanding of the inflammatory reaction is warranted and analysis of several proteins involved in the immune system may provide important insights. Obviously, studies appropriately addressing the gut-kidney crosstalk and specifically measure endotoxin levels [123], are needed to test our hypothesis about endotoxemia. The present manuscript considers kidney injury over a 2–5 month perspective, but further studies need to be conducted at different time scales such as cross-shift, cross-weekend, cross-harvest, and over several harvests as the immune system is highly dynamic. Qualitative studies investigating popular beliefs about ‘fever’ may improve our understanding of etiology and of what workers find are appropriate measures to take when feeling feverish.

Efforts should go into measuring heat stress and physical activity, and responses such as vasoactive and fluid-regulatory hormones among workers at risk of CKDnt, to improve translation of findings between occupational and experimental settings. One recent experimental study [124] designed to match conditions experienced by sugarcane workers found evidence of kidney injury associated with both dehydration and hyperthermia, illustrating how laboratory-to-workplace translational studies might inform workplace interventions. While experimental studies of heat/exercise-induced AKI in humans may be ethical, studies designed to produce CKDnt are not. Studies of this endpoint thus necessitate identification of at-risk populations, assessment of their working conditions and well-implemented interventions to improve these, and longitudinal follow-up. 

### 5.2. Implications for Workplaces and Intervention Studies

The theoretical framework developed was designed to help identify factors to address in workplaces and to further occupational intervention studies. It is unlikely that addressing a single one of these factors will prevent CKDnt, rather, several factors need to be considered simultaneously. 

1. Hydration and Hyperthermia: Hydration alone does not prevent heat stroke [72,73] and experimental heat/exercise-induced kidney injury occurs also in those well hydrated [124]. This is consistent with weak associations between overall liquid intake volumes and kidney injury in this and other sugarcane worker studies [6,9,14,115]. Conversely, there is a clear association with those jobs requiring the highest workload and subsequent highest core temperatures and kidney injury [6,8]. Adopting preventive measures limiting hyperthermia such as imposing rest schedules and limiting work at the hottest hours appears warranted, and active cooling may be considered [125]. 

2. Sugar and Electrolyte Consumption: With evidence on the adverse impact of sugary drinks on kidney health, workers should be recommended not to rehydrate with such drinks. Attention should be directed to providing the necessary nutrients needed for their job in other affordable ways. While dietary sources of electrolytes are the basis, supplying electrolyte solutions may be beneficial but the composition of such solutions needs to be specifically studied and dietary sources of electrolytes should not be overlooked. 

3. Ergonomics and Pain Relief: While workplace improvements ideally should prevent muscular damage and reduce the need for analgesics, workers should be informed of negative effects of NSAIDs. 

4. Healthcare and Rest: Preventive efforts to reduce kidney injury in this population should consider encouraging and enabling workers to seek healthcare and abstain from physically demanding work in heat when feeling unwell without this leading to personal economic disaster. This requires companies to have evaluation capacity and to provide workers with paid sick leave if there are signs of kidney injury or if they are sick with a febrile infection. Generally increasing weekly rest periods might improve kidney inflammation recovery and prevent long-term injury and should be considered in future intervention studies.

While workplace improvements can be adequately designed with attention paid to extant data, the implementation of such interventions should be carefully monitored and evaluated to assure protective and precautionary measures are being followed [8].

## Figures and Tables

**Figure 1 nutrients-12-01639-f001:**
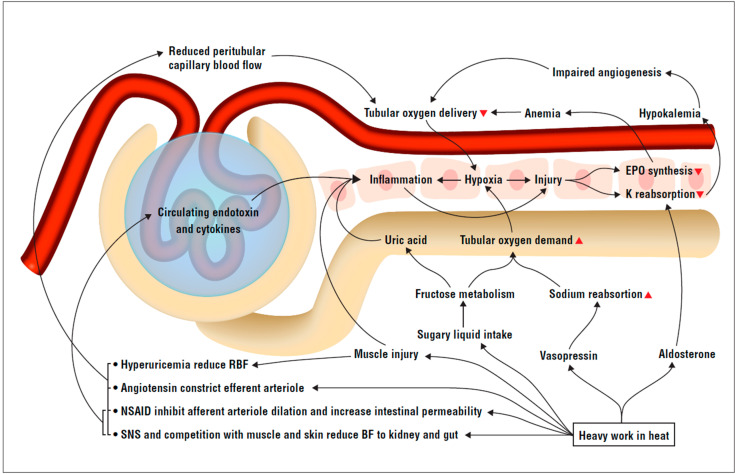
Schematic summary of potential pro-inflammatory stimuli in the sugarcane worker nephron. Red = blood vessels. Yellow = tubular lumen. White boxes = tubulointerstitial cells. RBF = renal blood flow. NSAID = non-steroidal anti-inflammatory drug. SNS = sympathetic nervous system. BF = blood flow. RAAS = renin–angiotensin–aldosterone system. EPO = erythropoietin. Heavy work in heat leads to loss of volume and electrolytes triggering RAAS activation and vasopressin release, reducing renal blood flow, as well as increasing sodium absorption and potassium excretion. Heavy exercise and work decreases renal and gut blood flow through direct competition over the cardiac output and through sympathetic neural pathways. NSAID used to treat musculoskeletal pain from heavy work inhibits afferent arteriolar dilation, further decreasing renal blood flow, while also increasing intestinal permeability. Increased gut permeability enables endotoxins to enter the blood stream and trigger cytokine release systemically and in the tubuli, activating an inflammatory response. This inflammatory response in the tubuli can be further promoted by muscle cell breakdown products such as uric acid and tubular fructose metabolism, which also lead to uric acid production. Inflammation and relative hypoxia in the tubuli cause tubular cell injury, leading to further loss of potassium and decreased EPO synthesis, which in turn further impair tubular oxygen delivery.

**Table 1 nutrients-12-01639-t001:** Associations between incident kidney injury (IKI) and categorical risk factors among male sugarcane cutters.

	Nicaragua (Adelante Cohort)				El Salvador (IEA Cohort)			
	Worker-Harvests				Worker Follow-up Occasions			
	Total	IKI	Incidence Ratio (IR) (95% CI)	IR Adjusted for Baseline eGFR (95% CI)	Total	IKI	Odds Ratio (OR) (95% CI)	OR Adjusted for Baseline eGFR (95% CI)
Age (years)								
18–30	320	32 (10%)	ref	Ref	289	22 (8%)	Ref	Ref
31–40	155	16 (10%)	1.0 (0.6–1.9)	0.9 (0.5–1.6)	179	32 (18%)	5.7 (1.3–25)	1.1 (0.3–3.8)
41–50	43	5 (12%)	1.2 (0.5–3.0)	0.9 (0.3–2.3)	108	20 (19%)	6.2 (1.1–33)	0.6 (0.1–2.4)
>50	14	0 (0%)	NA	NA	108	27 (25%)	17.5 (3.2–95)	0.8 (0.2–3.1)
eGFR, baseline (mL/min/1.73 m^2^)								
>90	358	29 (8%)	Ref	Not meaningful	398	16 (4%)	Ref	Not meaningful
90–60	163	23 (14%)	1.7 (1.0–3.0)		136	15 (11%)	5.0 (1.4–18)	
45–60	10	1 (10%)	1.2 (0.2–9.1)		57	19 (33%)	43 (8.6–211)	
<45	1	0 (0%)	NA		85	51 (60%)	290 (51–1663)	
CRP, baseline (mg/L)								
<3	423	40 (9%)	Ref	Ref	482	69 (14%)	Ref	Ref
3–10	85	12 (14%)	1.5 (0.8–2.8)	1.4 (0.7–2.6)	153	21 (14%)	1.2 (0.3–4.7)	0.6 (0.2–1.7)
10–20	12	0 (0%)	NA	NA	28	10 (36%)	22 (1.3–384)	0.9 (0.1–5.8)
>20	12	1 (8%)	0.9 (0.1–6.4)	0.7 (0.1–5.4)	13	1 (8%)	0.3 (0.01–62)	0.1 (0.01–3.5)
CRP, follow-up (mg/L)								
<3	340	10 (3%)	Ref	Ref	434	25 (6%)	Ref	Ref
3–10	142	14 (10%)	3.4 (1.5–7.5)	3.3 (1.5–7.5)	173	37 (21%)	8.7 (3.3–23)	5.0 (1.9–12.9)
10–20	24	9 (38%)	13 (5–31)	13 (5.1–31)	43	21 (49%)	58 (14–249)	21 (5.3–82)
>20	26	20 (77%)	26 (12–56)	25 (12–54)	34	18 (53%)	71 (15–337)	19 (4.7–76)
NSAID use at least once per week †								
No	444	37 (8%)	Ref	Ref	647	92 (14%)	Ref	Ref
Yes	88	16 (18%)	2.2 (1.2–3.9)	2.1 (1.2–3.8)	25	7 (28%)	2.9 (0.5–16)	2.8 (0.6–14)
Incident fever in past week(s) ††								
No	463	40 (9%)	Ref	Ref	384	41 (11%)	Ref	Ref
Yes	43	11 (26%)	3.0 (1.5–5.8)	3.1 (1.6–6.1)	287	58 (20%)	5.6 (1.9–16)	2.1 (0.9–5.0)
Sugary drink intake (L)					Not available			
<0.2	64	2 (3%)	Ref	Ref				
0.2–1	312	30 (10%)	3.1 (0.7–13)	3.1 (0.7–13)				
>1	155	21 (14%)	4.3 (1.0–18)	4.4 (1.0–19)				
Morning boli intake (N of 300 mL sachets)								
0	125	18 (14%)	Ref	Ref				
0–3	214	23 (11%)	0.7 (0.4–1.4)	0.8 (0.4–1.5)				
≥3	180	12 (7%)	0.4 (0.2–0.9)	0.5 (0.2–0.9)				
Morning water intake (L)								
0–2	76	12 (16%)	Ref	Ref				
2–5	209	18 (9%)	0.5 (0.3–1.1)	0.6 (0.3–1.2)				
>5	246	23 (9%)	0.6 (0.3–1.2)	0.6 (0.3–1.3)				
Mg, baseline (mmol/L)								
≥0.7	Not collected				455	63 (14%)	Ref	Ref
<0.7					67	29 (43%)	22 (3.7–124)	1.6 (0.4–6.1)
K, baseline (mmol/L)								
≥3.5	Not collected				496	82 (17%)	Ref	Ref
<3.5					26	10 (38%)	11 (0.9–138)	5.2 (0.7–38)
K, follow-up (mmol/L)								
≥3.5	Not collected				482	71 (15%)	Ref	Ref
<3.5					47	21 (45%)	13 (3.1–57)	7.2 (2.1–25)
Na, follow-up (mmol/L)								
≥137	283	20 (7%)	Not possible		153	9 (6%)	Not possible	
<137	15	1 (7%)			1	0 (0%)		
Incident biochemical changes *							eGFR coefficient (ml/min/1.73 m^2^) compared to baseline	
Hemoglobin <125 g/L								
No	Not collected				138	3 (2%)	Ref	
Yes					16	3 (19%)	−9 (−12, −5)	
Dipstick hematuria								
No	Not collected				105	2 (2%)	Ref	
Yes					16	3 (19%)	−5 (−8, −1)	
Dipstick proteinuria								
No	Not collected				141	5 (4%)	Ref	
Yes					9	4 (44%)	−10 (−15, −5)	
Uric acid crystals								
No	Not collected				142	7 (5%)	Ref	
Yes					10	2 (20%)	−6 (−11, −1)	
Non-hyaline cylinders								
No	Not collected				128	6 (5%)	Ref	
Yes					9	2 (22%)	−7 (−13, −1)	
Microscopy leukocyturia								
<25/fov	Not collected				146	4 (3%)	Ref	
≥25/fov					7	5 (71%)	−26 (−31, −21)	

† In the past two weeks in El Salvador. Missing observations as not all workers have questionnaire data from all visits, year 2 El Salvador. †† One week in year 1 Nicaragua, otherwise two weeks. Not excluding workers with fever at baseline in El Salvador as fever was not asked for. Missing observations as not all workers have questionnaire data from all visits, year 2 El Salvador. * Not present at baseline, excludes workers with the abnormality at baseline or missing data at baseline or follow-up. fov = field of view. Boli = 300 mL electrolyte solution bags containing 7 g of sugar, 50 mg sodium chloride, and 20 mg potassium monophosphate per 100 mL. Cross-harvest: from beginning of harvest (November) until end of harvest (April). Adelante cohort workers may be included in both years [8]. Cross-test interval: From beginning of harvest (November) until one of up to three follow-up visits. IEA cohort workers may be included in both years and have multiple cross-test intervals within one year ([15] and Supplement 1).

**Table 2 nutrients-12-01639-t002:** Association between incident kidney injury and continuous biochemical parameters in male sugarcane cutters.

Country	Year	Biochemical parameter		Median (interquartile Range)			Regression coefficients (95% CI)	
El Salvador †				Baseline	During harvest			
					Non IKI day	IKI day	IKI day vs. non IKI day regression coefficient	
	1	Worker-visits		74	145	9	Unadjusted	eGFR-adjusted
		WBC	10^9^/L	8.2 (7.2–9.0)	7.7 (6.6–8.5)	9.5 (9.3–10)	2.2 (1.3–3.1)	1.6 (0.6–2.7)
		Neutrophils		4.7 (3.9–5.5)	3.9 (3.3–4.5)	5.5 (5.3–5.8)	2.0 (1.3–2.8)	1.7 (0.8–2.6)
		Lymphocytes		2.6 (2.2–3.0)	2.9 (2.5–3.2)	3.0 (2.7–3.3)	0.1 (–0.3–0.4)	0.1 (–0.4–0.5)
		Monocytes		0.7 (0.6–0.9)	0.7 (0.5–1.0)	0.9 (0.8–1.0)	0.2 (–0.1–0.5)	0.0 (–0.4–0.3)
		Urine–pH	–	6 (5.5–6.5)	6 (5.5–6.5)	5.5 (5.5–6.0)	–0.4 (–0.8–0.0)	–0.3 (–0.7–0.2)
		Hemoglobin	g/L	148 (143–153)	142 (135–148)	125 (123–130)	–9 (–16––2)	3 (–4–9)
		Sodium	mmol/L	141 (139–141)	140 (139–141)	140 (139–141)	0 (–1–1)	0 (–1–1)
	1 + 2	Worker–visits		345	580	101		
		CRP	mg/L	1.4 (0.7–3.3)	1.5 (0.7–4.0)	7.5 (3.0–13)	7.6 (5.7–9.5)	3.3 (1.0–5.7)
		Uric acid	mmol/L	361 (310–443)	328 (281–389)	586 (477–672)	117 (98–135)	32 (14–50)
		CPK	µkat/L	2.4 (1.8–3.4)	2.8 (2.2–3.9)	3.8 (2.6–5.1)	0.8 (0.3–1.4)	0.7 (0.0–1.3)
	2	Worker–visits		213	429	92		
		Potassium	mmol/L	3.9 (3.5–4.1)	3.9 (3.5–4.2)	3.4 (3.0–3.9)	–0.3 (–0.5––0.2)	–0.1 (–0.3–0)
Nicaragua ††				Baseline	At end–harvest			
					No IKI	IKI	Cross–harvest trend difference between IKI and non–IKI workers	
	1 + 2	Workers		533	480	53	Unadjusted	eGFR–adjusted
		CRP	mg/L	1.1 (LoQ–2.3)	1.6 (0.8–3.7)	12 (5.8–24)	15 (12–17)	12 (9–14)
		Uric acid	mmol/L	324 (277–374)	326 (282–378)	428 (345–499)	61 (47–74)	–5 (–19–9)
		CPK	µkat/L	2.6 (2.0–3.4)	3.5 (2.6–4.8)	3.4 (2.6–5.2)	0.3 (–0.2–0.8)	0.0 (–0.6–0.6)

All models were linear mixed models, except CRP in Nicaragua which was modeled using mixed effects tobit regression as results below 0.6 were not quantified. Adjustment for eGFR is for eGFR at the follow-up date. † El Salvador analyses are by worker-visit units, meaning the same individual may have been sampled at up to three occasions during the same harvest, and comparisons in the regression models are between visits during which there was or was not concomitant IKI within that same individual and other individuals, as modeled using a mixed effects model with random effects for individuals. †† Nicaragua analyses are by worker-harvest units as workers were only sampled at baseline and at end of harvest, again modeled with a random effects structure for individuals to account for the repeated measurements within the same worker. CI = confidence interval, WBC = white blood cells, CPK = creatine phosphokinase, CRP = C-reactive protein, LoQ = limit of quantification, for CRP in Nicaragua 0.6 mg/dL, IKI = incident kidney injury, defined as ≥0.3 mg/dL or ≥50% increase in serum creatinine since baseline.

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
