# Peer review of "Pathophysiological Mechanisms by which Heat Stress Potentially Induces Kidney Inflammation and Chronic Kidney Disease in Sugarcane Workers"

_nutrients, 2020, doi:10.3390/nu12061639_

Round 1
Reviewer 1 Report
Autors presented a manuscript about Mesoamerican nephropathy and potential epidemiological factors involved in it. Manuscript has a strange structure, which links review + original paper elements. Despite the subject of Mesoamerican nephropathy is very important and need to be carefully studied (what Authors of this paper did many times even recently:
1)Wesseling C, Glaser J, Rodríguez-Guzmán J, Weiss I, Lucas R, Peraza S, da Silva AS, Hansson E, Johnson RJ, Hogstedt C, Wegman DH, Jakobsson K. Chronic kidney disease of non-traditional origin in Mesoamerica: a disease primarily driven by occupational heat stress. Rev Panam Salud Publica. 2020 Jan 27;44:e15. doi: 10.26633/RPSP.2020.15
2) Hansson E, Glaser J, Weiss I, Ekström U, Apelqvist J, Hogstedt C, Peraza S, Lucas R, Jakobsson K, Wesseling C, Wegman DH. Workload and cross-harvest kidney injury in a Nicaraguan sugarcane worker cohort. Occup Environ Med. 2019 Nov;76(11):818-826. doi: 10.1136/oemed-2019-105986
3) Glaser J et al. Preventing kidney injury among sugarcane workers:
promising evidence from enhanced
workplace interventions. Occup Environ Med 2020;0:1–8. doi:10.1136/oemed-2020-106406)
this manuscript needs some improvement:
- Manuscript does not have an abstract;
- Please improve the quality of Figure 1;
- if possible please make introduction & theoretical framework shorter, providing to many information makes this manuscript look like a review;
- do not write about your suggestions/beliefs (in the The theoretical framework writing about beneficial effects of physical exercise is obvious and not necessary);
- Please improve Table 1, try to collect data in narrow areas or use lines to seperate results, sometimes they in the middle of the page and Reader may have a problem with guessing which results shows what (pages 9-10), table 2 looks much better in this manner;
- authors use term sympathetic nervous system (SNS) - page 3, but later not: "Activation of the sympathetic system decrease RBF." (page 5), please terms/abbreviations in the same manner;
- there are too many, unresolved observational results; positive urine dipstick tests for protein or leucocytes need to be further verified if patients have urinary tract infection, especially in the presence of symptoms, which may exacerbate kidney damage caused by heat; also non specific symptoms or different symptoms reported by patients related with the same pathology makes analyzing of such results very hard. Onset of symptoms fever, dysuria) is also crucial, did they also occur before heat-kidney damage?
Author Response
Reviewer 1
- Manuscript does not have an abstract;
The manuscript was uploaded as instructed in the submissions system with an abstract but it unfortunately does not seem to have been included in the document created by the journal. It is now included in the resubmitted document.
- Please improve the quality of Figure 1;
We have consulted with a graphic designer to improve the quality of Figure 1.
- if possible please make introduction & theoretical framework shorter, providing to many information makes this manuscript look like a review;
We have reviewed the introduction and theoretical framework and removed unnecessary information (several changes on pages 2-6). Since the kidney is a complex organ our framework has been designed to cover the range of risk factors and mechanisms which may have an unfavorable effect on kidney health rather than focusing on one factor. In this way we do not reduce our analysis to one single pathway or overly simplified pathways. Especially considering that the pathophysiology of chronic kidney disease not explained by known factors is uncertain we consider it crucial to help readers consider several coinciding contributing mechanisms. As the second reviewer considered we did a “great job” describing the theoretical framework we have focused on tightening the text rather than simplifying it.
- do not write about your suggestions/beliefs (in the The theoretical framework writing about beneficial effects of physical exercise is obvious and not necessary);
We agree - this sentence has been removed (line 18, page 4).
- Please improve Table 1, try to collect data in narrow areas or use lines to seperate results, sometimes they in the middle of the page and Reader may have a problem with guessing which results shows what (pages 9-10), table 2 looks much better in this manner;
We agree and have revised this table, as suggested, we have now used lines and also shading to help the reader navigate this table.
- authors use term sympathetic nervous system (SNS) - page 3, but later not: "Activation of the sympathetic system decrease RBF." (page 5), please terms/abbreviations in the same manner;
We introduce the SNS abbreviation and now use that throughout in the text. (page 5, lines 31-33).
- there are too many, unresolved observational results; positive urine dipstick tests for protein or leucocytes need to be further verified if patients have urinary tract infection, especially in the presence of symptoms, which may exacerbate kidney damage caused by heat; also non specific symptoms or different symptoms reported by patients related with the same pathology makes analyzing of such results very hard. Onset of symptoms fever, dysuria) is also crucial, did they also occur before heat-kidney damage?
We note (page 13, line 14) that there is no urine dipstick which is positive for nitrite, making it unlikely that UTI by nitrite-producing bacteria is a major cause of the few instances of protein- and leukocyturia. While we acknowledge that we cannot rule in or out other causes of infection, we noted (page 13, line 17) other studies have ruled out UTIs through cultures. Dysuria was an uncommon complaint among those with kidney injury. Due to a lack of space we cannot present data analysing all symptoms that have been collected, but focused on fever as that was the symptom most obviously indicative of inflammation.
Fever assessed at baseline was not associated with reduced kidney function at baseline or increased risk of kidney injury during harvest, but, as reported, fever assessed at the end of harvest was associated with an increased risk of kidney injury during harvest.
We consider that we cannot, by design, determine the exact temporal relationship between symptoms onset and kidney damage - that would require much more regular questionnaires on symptoms and kidney health check-ups, and it is unlikely that this would be acceptable to workers. We note that workers who report fever within the 2 weeks before assessment were substantially more likely to have kidney damage. Which came first is not possible to answer. Nonetheless, we consider the association between kidney injury and a symptom that is indicative of inflammation interesting as it suggests inflammatory pathways mediate kidney damage. An associated symptom also might enable early identification and underline the importance of not working when feeling unwell irrespectively of whether fever or kidney injury occurred first.
We have moved the methods section that previously was in a supplement to the main manuscript to clarify what was collected at each point in time.
Reviewer 2 Report
The authors do a great job describing a theoretical framework for kidney injury in sugarcane workers and support this framework with epi-level data. I have a few comments to help make the manuscript clearer that I hope the authors will find the comments below helpful.
-Given the relationship between muscle mass and breakdown on serum creatinine, the use of cystatin-c to define AKI may offer a clearer understanding of this data. Do the authors have the ability to measure cystatin-c in their own cohorts/data?
-Can the authors explain why they used the pre-shift samples in the IEA cohort rather than the post shift? It would make sense that the post-shift samples would be more reflective of the injury that took place “after” the strenuous physical activity. This likely biases the results towards the null, and associations may be even stronger using post-shift samples.
-Table 1 is visually hard to follow, and it may be better presented as a figure, maybe a forest plot, to easily highlight significant associations. Separate plots will be needed for ORs and B-coefficients.
-Unclear why the authors only choose to adjust for baseline eGFR, as there are many other confounders. It would be helpful to do a multivariable assessment including all the significant exposures/ or exposures of interest to fully understand the contribution of each to the model.
-how were the exposure variable cut-offs determined in Table 1? It does not appear to be a tertile/quartile cut off as each category has significantly different sample size. Can the authors justify the reason for these cut-offs, were they all a priori hypotheses? If not, it may be better to present spline plots in the supplementary to justify the use of the cut-offs presented or to simply use the continuous variable if the relationships are linear in nature.
- The authors need to clarify which time point were chosen for the exposures and outcomes in table 1 and table 2 and how they were analyzed in the main manuscript (not just the supplementary) as these points are key to making the correct temporal associations between exposure and outcome.
Author Response
Reviewer 2
The authors do a great job describing a theoretical framework for kidney injury in sugarcane workers and support this framework with epi-level data. I have a few comments to help make the manuscript clearer that I hope the authors will find the comments below helpful.
Thank you.
-Given the relationship between muscle mass and breakdown on serum creatinine, the use of cystatin-c to define AKI may offer a clearer understanding of this data. Do the authors have the ability to measure cystatin-c in their own cohorts/data?
We have plans to report our Cystatin C measurements but note these estimates give a similar picture as the creatinine-based estimates. We are in the process of analyzing these data and writing up a separate paper on that as it answers important questions and also raises some additional interesting issues.
-Can the authors explain why they used the pre-shift samples in the IEA cohort rather than the post shift? It would make sense that the post-shift samples would be more reflective of the injury that took place “after” the strenuous physical activity. This likely biases the results towards the null, and associations may be even stronger using post-shift samples.
We appreciate your concern with the possible bias to the null that we accepted in focusing on only the pre-shift samples. One reason was that there were no cross-shift data for the Nicaraguan cohort. Post-shift measurements likely reflect a combination of injury accumulated under a longer cross-harvest perspective and a more short-term cross-shift perspective that would need to be accounted for in the analysis. We chose to use only the pre-shift measurement for El Salvador to avoid the much more complex analysis and likely confusing for the reader if we included both pre- and post-shift measurements for only one of the cohorts.
Consistent with the reviewer’s concern, we have recently collected additional cross-shift and post-weekend-rest samples combined with a range of work-site physiological measurements and plan to revisit the important question of what happens during the shift and after in publication of that work. For the current manuscript we considered the additional complexity of introducing the cross-shift perspective too great and likely distracting.
We have clarified that our study is limited to the cross-harvest perspective, and that further studies need to be made to look at other time perspectives (page 15, lines 133-134).
-Table 1 is visually hard to follow, and it may be better presented as a figure, maybe a forest plot, to easily highlight significant associations. Separate plots will be needed for ORs and B-coefficients.
We agree and have revised this table as noted above. We explored the possibility of replacing the table with forest plots but trying this out it became evident that some of the effect estimates (such as for CRP) are very high, making it difficult to discern estimates for more moderate, but still important risk factors. We hope instead that the revised table is more readable.
-Unclear why the authors only choose to adjust for baseline eGFR, as there are many other confounders. It would be helpful to do a multivariable assessment including all the significant exposures/ or exposures of interest to fully understand the contribution of each to the model.
Despite being one of the largest occupational CKDnt cohort studies to date there are not enough outcome events to include other factors in a single regression model. In order to avoid the risk of violating regression modelling assumptions we could not include all significant exposures in the same model. We have however run some targeted analyses that are now included in a supplement and very briefly mentioned in the methods section.
- adjusting for harvest (harvest 1 and 2) and interaction with occupation (seed and burned cane) for each of the other variables one at a time to see if there is confounding with the improved intervention (Glaser et al 2020). Conclusion: overall results do not change substantially though there was very slight attenuation of NSAID and protective liquid intake estimates consistent, with more rest in harvest 2 where there were fewer reports of NSAID use, more liquid intake, and better outcomes.
- Adjusting for protective liquid types (adding both water and electrolyte intake to model). Conclusion: Estimates for electrolyte solution are not changed, but there was more of a U-shaped association for water intake, with >5L at 0.9, potentially suggesting electrolyte intake may potentially be more important.
- Including combinations of NSAID, fever and CRP in the same model to see if NSAID treatment of inflammation confounds the association between NSAIDs and IKI. Conclusion: There is an effect of NSAIDs independent of inflammatory markers or symptoms which is not reduced by adjustment for fever or CRP. If fever and CRP enter the model simultaneously the effect of fever is largely abolished, which makes sense as they both represent inflammation with CRP measuring this better in these data.
-how were the exposure variable cut-offs determined in Table 1? It does not appear to be a tertile/quartile cut off as each category has significantly different sample size. Can the authors justify the reason for these cut-offs, were they all a priori hypotheses? If not, it may be better to present spline plots in the supplementary to justify the use of the cut-offs presented or to simply use the continuous variable if the relationships are linear in nature.
We have added the rationale behind the cut-offs to page 7, section “3.1.1.4 Covariate definitions”
Use of dichotomised electrolyte values at the reference range as normal-abnormal is motivated by what is usually considered pathological in the clinic, in which the exact value within the reference range is often not given much attention. Having an abnormal value is considered indicative of a disequilibrium. As an example, a difference in potassium concentration of 0.5 between 4.3 and 3.8 is not a cause of much concern, while the 0.5 difference between 3.8 and 3.3 is more so.
However, we agree that analyses on a continuous scale may be interesting from a physiological rather than clinical perspective, and both these perspectives are important for this disease. We thus include such analyses in Table 2 for potassium, sodium and hemoglobin for which it is possible (magnesium was only measured at baseline). This however does not change our conclusions as it shows decreasing potassium and hemoglobin concentrations while stable sodium concentrations associated with kidney injury, just as found for the dichotomised exposures. Urine dipstick test results are categorical and could not be analysed as continuous.
Due to the questionnaire structure (using pictures the workers were asked to recall many containers of different sizes and types of fluid they had consumed in the previous day), the liquid intake data is not strictly continuous but rather semi-categorical and therefore not appropriately analysed as continuous.
- The authors need to clarify which time point were chosen for the exposures and outcomes in table 1 and table 2 and how they were analyzed in the main manuscript (not just the supplementary) as these points are key to making the correct temporal associations between exposure and outcome.
We agree and have moved the entire methods section from the supplementary material to the main manuscript (page 6-8). Some sentences in the 3.1.1 section have therefore been dropped/modified.
Round 2
Reviewer 1 Report
Authors adequately answered to all my questions.
Reviewer 2 Report
No further comments. Authors addressed points sufficiently.